

# Effect of temperatures on some biological parameters of *Aphis craccivora* Koch (Hemiptera: Aphididae) on lentil

Muhlis Sezgin[1], Merve Akyıldız[1], Selime Olmez Bayhan[2] and Erol Bayhan[2]

[1] Turkish Ministry of Agriculture and Forestry General Directorate of Agricultural Research and Policies, Diyarbakır, Turkey
[2] Dicle (Tirgris) University, Diyarbakır, Turkey

## ABSTRACT

*Aphis craccivora* Koch (Hemiptera: Aphididae) is one of the aphid species known to cause economically important damage in lentil fields. Temperature effects on the developmental period, age stage, stage-specific survival, and reproductive capacity of *A. craccivora* were studied on lentil under laboratory conditions at four fixed temperatures (22.5, 25, 27.5, and 30 °C). As a result of the study, the developmental periods of nymphal stages of *A. craccivora* were determined as 26.04, 17.24, 23.98 and 26.74 degree days, respectively. *A. craccivora* had the longest pre-adult developmental period at 22.5 °C, the shortest developmental period at 27.5 °C, the highest productivity value was found as 62.74 at 25 °C, and the lowest value was found as 11.28 at 30 °C. In the study, the shortest preimaginal phases of development period, the longest individual lifespan, and the highest productivity were observed at 25 °C. In the study, the highest intrinsic rate of increase ($r$) of *A. craccivora* was 0.36 (day$^{-1}$), the net reproductive rate ($R_0$) was 63.67 (female/female), finite rate of increase ($\lambda$) was 1.43 day$^{-1}$ and gross reproductive rate (GRR) was 80.88 at 25 °C. The shortest mean generation time (T) value was 8.63 days at 30 °C, while the shortest population doubling time (DT) was observed at 25 °C with 1.93 days. The most suitable temperature for population development was 25 °C for *A. craccivora*. The role of temperature as a key factor in determining the *A. craccivora* population on lentil is discussed.

## INTRODUCTION

Aphids (Hemiptera: Aphididae) are pests that can cause significant problems in many plants. In temperate climates, aphids reproduce asexually (parthenogenetically) throughout the year. Equipped with piercing-sucking mouthparts, both nymph and adult aphids feed by puncturing and sucking plant sap from leaves, shoots, and stems. As aphids feed, they excrete a sugary substance called honeydew, which coats the plants and promotes the growth of saprophytic fungi, leading to sooty mold (fumagine). Additionally, aphids are among the most important vectors of plant viral diseases. Under high aphid

Corresponding author
Muhlis Sezgin,
muhlis.sezgin@tarimorman.gov.tr

infestation, plants may exhibit stunted growth, reduced yield, and diminished crop quality (*ZMTT, 2008*).

Pests also play a significant role in the low yield and quality of legumes, particularly in lentils. The lentil (*Lens culinaris*, Fabales: Fabaceae), one of the host plants of aphids, is a crop that can grow in a wide pH range (5.5–9) and adapt to different soil types. In today's world, where global warming is a widely discussed issue, lentils are recognized for their drought resistance and serve as a valuable food source in many countries. They also play an important role in sustainable agricultural systems (*Ramirez & Cantero, 2024*). Lentils, which thrive in arid and semi-arid regions, possess various adaptation strategies, including escape, avoidance, and tolerance mechanisms, to counteract the adverse effects of drought (*Al Noor et al., 2024*). Additionally, lentils are a highly nutritious food, rich in essential components such as fiber, minerals, proteins, and amino acids. For this reason, they are a crucial dietary staple in low- and middle-income countries (*Alexander et al., 2024*).

Among aphid species known to cause economically significant damage in lentil fields are *Acyrtosiphon pisum* (Harris) (Hemiptera: Aphididae) and *A. craccivora* Koch, 1854 (Hemiptera: Aphididae) (*Muehlbauer, Cubero & Summerfield, 1985*). Aphids typically infest plants during their early growth stages, inhabiting terminal leaves and shoots. Through the toxic substances they introduce into the plant system, they cause deformations such as leaf curling, wrinkling, and discoloration. In red lentil production, this situation can lead to yield losses, posing a substantial economic problem (*ZMTT, 2008*).

Since there is a relationship between temperature and the developmental duration of pest insects, degree-day modeling serves as an important predictive method for pest management. The objective is to forecast the emergence period of pests and determine the optimal timing for control measures. Degree-day models are categorized into two groups: simple and advanced. The simple degree-day model is based on a linear function between organism development rate and temperature, whereas the advanced degree-day model relies on the principle of effective temperature, which accounts for the specific temperature threshold beyond which insect development progresses (*Birgücü & Karsauran, 2009*). These models are particularly crucial components of integrated pest management (IPM) systems in agriculture. Given that aphids possess a high reproductive potential under favorable environmental conditions, understanding their biology is essential for effective pest control strategies.

In this study, the development of *A. craccivora*, which is a problem in red lentil cultivation areas in the Southeastern Anatolia Region, on lentil plants under different temperature conditions in the laboratory was investigated. Some biological characteristics contributing to pest control and thermal constant values considered in determining the time of spraying in the control were revealed. This article reports the effect of temperature on development time, reproduction rate and mortality of *A. craccivora*, to better understand their population dynamics on lentil in Turkey.

## MATERIALS AND METHODS

### Plant source

The Fırat87 red lentil variety was used in the study. Lentil plants were grown in a fully automated greenhouse environment with $24 \pm 2$ °C temperature, $40 \pm 5\%$ humidity, and 16:8 L: :D conditions. 21-liter pots with a diameter of 40 cm and a height of 31 cm, containing a 1/2 ratio of soil and peat, were used for lentil production. The lentil plants in the pots where the seeds were planted were watered regularly at 72-h intervals, and the plants were ready for use 3 weeks after seed emergence.

The Fırat 87 was bred in 1987 by Diyarbakır GAP International Agricultural Research and Education Center and registered in 2012. It has a plant height of 40–45 cm, first pod height of 16–20 cm, flowering days of 162–167 days, semi-recumbent growth form, red cotyledons, and is a winter-resistant variety. Based on its pink, black-dotted shell color, morphological characteristics, and general red lentil chemical composition, Fırat87 lentil variety is thought to contain important phytochemicals such as phenolic compounds (especially the shell part containing anthocyanins and proanthocyanidins), dietary fiber, saponins, phytosterols, and phytic acid. These compounds contribute to the potential benefits of Fırat87 lentils, such as antioxidant, anti-inflammatory, and cardiovascular health (*Şahin, 2021*).

### Insect source

Aphid individuals found on randomly selected lentil plants in the lentil cultivation areas of Sur district of Diyarbakır province were obtained by shaking the plants into a plastic container. The aphids obtained from the field were transferred to lentil plants grown in pots in a climate room with a temperature of $25 \pm 1$ °C, $60 \pm 5\%$ humidity, and 16:8 L: D conditions. The adult individuals obtained from this stock culture were placed as one individual each in 6 cm diameter plastic petri dishes (test unit) containing lentil plant shoots cut into 3 cm lengths and taken to separate climate chambers with four different temperatures, $65 \pm 5\%$ humidity, and 16:8 L: D conditions for production. Here, one progeny was obtained from an aphid in climate cabinets with different temperatures and $65 \pm 5\%$ humidity and 16:8 L: D conditions. Thus, studies were conducted on the $F_1$ progeny obtained (*Bayhan et al., 2005a*, *2006*; *Aleosfoor & Fekrat, 2014*). In the study, 6 cm diameter plastic petri dishes were used for each individual. Three 0.5 cm diameter openings created on the lid of each petri dish to prevent ventilation and possible escapes were closed with silicone gauze. Then, filter paper was placed under the petri dishes, and these articles were fixed to create a test unit.

### Design of trials

#### Growth threshold-thermal constant trials

To determine the development of *A. craccivora* on red lentil at temperatures of 22.5, 25, 27.5, and 30 °C; for each temperature value, newborn aphid nymphs were placed in the previously mentioned test unit with the help of a soft-tipped brush. Filter papers fixed under the petri dishes were moistened daily, and these nymphs were fed lentil shoots cut

into 3 cm lengths every 24 h. The dates of newborn aphid nymphs, nymphal developmental stages according to the first change, developmental times, and mortality rates, and developmental time and survival rates were recorded by observing the individuals under a stereomicroscope every 24 h until they became adults and released offspring. Experiments were conducted with at least 50 replicates for each temperature (*Lu & Kuo, 2008*; *Aleosfoor & Fekrat, 2014*; *Bayındır & Birgücü, 2016*).

### Studies on some biological properties of Aphis craccivora at different temperatures

Newborn aphid nymphs from the F1 generations of *A. craccivora* obtained from the study were placed in the test unit with wet filter paper and lentil shoots cut into 3 cm lengths with the help of a soft-tipped brush, and the lid of the test unit was fixed using parafilm. The filter papers in the test units were moistened daily. Aphid nymphs were fed fresh shoots of lentil plants every 24 h. All experiments were carried out at four constant temperatures ranging from 22.5, 25, 27.5, and 30 ± 1 °C, 65 ± 5% humidity, and 16:8 L:D conditions. In all temperatures, preimaginal phases of development periods, molting, survival, and mortality rates of adults and newborn nymphs in the test units were counted under a stereomicroscope. After counting, the nymphs were removed from the test units, and their counts were recorded daily. Each experiment was conducted with at least 50 replicates for each temperature (*Ölmez Bayhan, Bayhan & Ulusoy, 2003*; *Aleosfoor & Fekrat, 2014*; *Bayındır & Birgücü, 2016*).

### Statistical analysis

Development of aphids under different temperature conditions in experiments. The linear model presented by *Campbell et al. (1974)* was used to calculate C = lower development threshold and K= thermal constant (total effective temperature required to complete a generation) for aphids. In this calculation;

in the formula d(T) = a + bT,

T = temperature (°C), d(T) = development rate (ratio of development time, 1/t), a and b parameters are constants. The development threshold was calculated according to the formula C = −a/b (°C) and the thermal constant was calculated according to the formula K = 1/b (degree days) (*Bayhan et al., 2005b*, *2006*; *Bayındır & Birgücü, 2016*; *Özgökçe, Bayındır & Karaca, 2016*). The values obtained at four different constant temperatures regarding the development times of pre-adult stages of aphids were analyzed using analysis of variance (ANOVA) (SPSS package 2026), and the differences between the treatments were determined by the Tukey test.

The demographic parameters of *A. craccivora* populations at four different temperatures were determined according to the methods described by *Birch (1948)*. Net reproductive rates (Ro), mean generation time (T), intrinsic increase rate ($r_m$), finite rate of increase (λ), and population doubling time (DT) were calculated. Raw data on developmental time, survival, longevity, and female fecundity were analyzed based on the age-stage, two-sex life table theory (*Chi & Liu, 1985*; *Chi, 1988*, *2022*) using the TWOSEX-MSChart computer program (*Chi, 2020*). Age-specific survival rate ($s_{xj}$) (x = age

and j = stage), age-specific survival rate ($l_x$), age-specific fecundity ($m_x$), and life table parameters (intrinsic increase rate ($r$); finite increase rate ($\lambda$); net reproductive rate ($R_0$); mean generation time (T); adult pre-oviposition period (APOP); and total pre-oviposition period (TPOP) were calculated accordingly. The intrinsic increase rate (r) was determined by iteratively solving the Euler–Lotka equation with age indexed from 0 (*Goodman, 1982*):

$$\sum_{x=0}^{\infty} e^{-(x+1)} lxmx = 1.$$

The finite growth rate ($\lambda$) and $R_0$ were calculated as follows:

$$\lambda = e^r.$$

$$R_0 = \sum_{x=0}^{\infty} l_x m_x.$$

The mean generation time (T) was then calculated using the following equation:

$$T = \frac{\ln R_0}{r}.$$

## RESULT AND DISCUSSION

In this study, data on the coefficient of determination ($r^2$), lower developmental threshold, and developmental period (DD/degree-days) for each nymphal stage of *A. craccivora* on red lentil plants under laboratory conditions at temperatures of 22.5, 25, 27.5, and 30 °C are presented in Table 1. Accordingly, the development periods for the nymphal stages of *A. craccivora* were determined to be 26.04, 17.24, 23.98, and 26.74 degree-days, respectively. The linear regression equations for each nymphal period show the relationship between temperature and development rate. Positive slope values indicate that the development rate increases as temperature rises. Notably, the $r^2$ value is quite high (ranging from 0.73 to 0.9784 and averaging 0.9084), which confirms the connection between temperature and development. The high $r^2$ values, especially in the third and fourth stages, suggest that development during these periods is heavily dependent on temperature. As the nymphal stages of *A. craccivora* progress, the lower development threshold value decreases and varies; the fourth stage has the lowest threshold, while the second stage has the highest. This may indicate that different temperatures are required at different nymphal stages. The fourth nymphal stage, with its low lower development threshold, also has the longest development time, whereas the second stage, with its high threshold, has the shortest development time (Table 1). Overall, the lower development threshold was 11.20, and the development time was calculated at 23.15 DD/degree-day. These data can help determine the most effective timing for pest control strategies.

In a similar study, *Dona & Satar (2024)* found the thermal constant (1/a) (°C days) values of *A. craccivora* nymphs on bean leaves at 16, 20, 24 and 28 °C under laboratory conditions as 7.69, 7.54, 20 and 6.67 days, respectively, and $r^2$ values between 0.730, 0.873, 0.222 and 0.769, respectively, and the very low $r^2$ value and high thermal constant value in

**Table 1 Regression equation and parameters of the development period rate of *Aphis craccivora* reared on lentil plants at four constant temperatures (22.5, 25, 27.5 and 30) under laboratory conditions.**

| İnstar | N | Regression equation | Mean ± Std. | R | $r^2$ | Lower developmental threshold (C) | Developmental time in DD (K) |
|---|---|---|---|---|---|---|---|
| 1 | 4 | Y = −0.4361 + 0.0384X | 0.57 ± 0.15 | 0.854[a] | 0.73 | 11.36 | 26.04 |
| 2 | 4 | Y = −0.8595 + 0.058X | 0.66 ± 0.21 | 0.875[a] | 0.765 | 14.82 | 17.24 |
| 3 | 4 | Y = −0.3964 + 0.0417X | 0.70 ± 0.14 | 0.980[a] | 0.961 | 9.51 | 23.98 |
| 4 | 4 | Y = −0.3017 + 0.0374X | 0.68 ± 0.12 | 0.989[a] | 0.978 | 8.07 | 26.74 |
| Total immature | 4 | Y = −0.484 + 0.0432X | 0.65 ± 0.15 | 0.953[a] | 0.908 | 11.20 | 23.15 |

**Note:**
The 'a' sign indicates that the statistical significance (or insignificance) of the R value is explained by the F-statistic and $p$-value given in this footnote (Instar 1 F:5.408 $p$ = 0.146, Instar 2 F:6.525 $p$ = 0.125, Instar 3 F:49.603 $p$ = 0.020, Instar 4 F:90.543 $p$ = 0.011, Total immature F:19.837 $p$ = 0.047).

the third nymphal stage showed a great difference. It is thought that the different results between the findings of this study and the data of *Dona & Satar (2024)* may be due to the different temperatures and nutrient use in the study.

In this study, the mean and standard error information of the data obtained from the experiments on development time, adult period, total lifespan, and productivity of *A. craccivora* on lentil plants under different temperature conditions are given in Table 2. The longest preimaginal phases of development period were obtained at 22.5 °C, while the shortest development period was at 27 °C. The longest average adult lifespan was 17.24 days at 25 °C, while the shortest average adult lifespan was 6.83 days at 30 °C. The differences in the lifespans of adults under different temperatures were statistically significant. The productivity values of *A. craccivora* on lentil plants under different temperatures are presented in Table 2. The highest productivity value was 62.74 at 25 °C, while this value was 23.76 at 27.5 °C. The lowest productivity values were 11.28 at 30 °C and 13.74 at 22.5 °C, which were statistically in the same group. Accordingly, the shortest pre-adult development time, longest individual life span, and highest productivity were observed at 25 °C.

Similar to this study, *Dona & Satar (2024)* investigated some biological properties of *A. craccivora* on pinto bean (*Phaseolus vulgaris* L.) at different temperatures and found that pre-adult development times were 1.4, 1.6, 1.1 and 1.7 d at 20.5 °C, 1.0, 1.1, 1.8 and 1.4 d at 24 °C and 1.1, 1.0, 1.5 and 1.3 d at 28 °C, respectively. Again, according to *Dona & Satar (2024)*, the lifespan of *A. craccivora* was determined as 22.2, 14.1, 12.4, and 10.8 d at 16, 20, 24, and 28 °C temperature conditions, respectively, whereas in this study, the highest life span was found at 25 °C with 17.24 d (Table 2). According to *Dona & Satar (2024)*, the number of offspring left by *A. craccivora* was 32.1, 40.8, 45.2, and 26.9 at 16, 20, 24, and 28 °C, respectively, whereas the highest productivity value was determined in this study with 62.74 at 25 °C (Table 2). Although the data on the development time of *A. craccivora* in both of these studies showed similar values, the values related to longevity and productivity differed. According to these results, it is thought that the values differ because both the temperature values considered in the studies and the foods used are different.

**Table 2 Developmental time (days, mean ± SE) of *Aphis craccivora* reared on lentil plant at four constant temperature.**

| | 22.5 | | 25 | | 27.5 | | 30 | |
|---|---|---|---|---|---|---|---|---|
| | $n$ | Time | $n$ | Time | $n$ | Time | $n$ | Time |
| 1st nymph period | 50 | $2.6 \pm 0.9^a$ | 54 | $1.85 \pm 0.1^b$ | 62 | $1.39 \pm 0.06^c$ | 60 | $1.55 \pm 0.1^{bc}$ |
| 2nd nymph period | 49 | $2.71 \pm 0.18^a$ | 54 | $1.59 \pm 0.08^b$ | 60 | $1.2 \pm 0.44^c$ | 57 | $1.35 \pm 0.08^{bc}$ |
| 3rd nymph period | 45 | $2.16 \pm 0.16^a$ | 54 | $1.48 \pm 0.08^b$ | 57 | $1.42 \pm 0.09^b$ | 51 | $1.41 \pm 0.09^b$ |
| 4th nymph period | 43 | $2.09 \pm 0.12^a$ | 54 | $1.63 \pm 0.12^b$ | 55 | $1.56 \pm 0.1^b$ | 42 | $1.67 \pm 0.12^b$ |
| Adult | 43 | $14.65 \pm 0.72^b$ | 54 | $17.24 \pm 0.75^a$ | 55 | $11.51 \pm 0.63^c$ | 42 | $6.83 \pm 0.54^d$ |
| Mean longevity | 50 | $21.74 \pm 1.11$ | 54 | $23.8 \pm 0.83$ | 62 | $15.56 \pm 0.78$ | 60 | $10.28 \pm 0.59$ |
| Fecundity (Mean) | 35 | $13.74 \pm 1.03^c$ | 50 | $62.74 \pm 2.85^a$ | 51 | $23.76 \pm 1.58^b$ | 39 | $11.28 \pm 1.58^c$ |

**Note:**
As demonstrated by the statistical data for the four constant temperature development periods of Aphis craccivora grown on lentil plants, the different letters a–d given in the rows are statistically different from each other since all $p$ values are less than 0.05 (Tukey's HSD; $p < 0.05$; $F_{first\ instar} = 28.551$, df = 3, $p = 0.000$, $F_{second\ instar} = 40.780$, df = 3, $p = 0.000$, $F_{third\ instar} = 9.301$, df = 3, $p = 0.000$, $F_{fourth\ instar} = 4.146$, df = 3, $p = 0.007$, $F_{adult} = 42.077$, df = 3, $p = 0.000$, $F_{fecundity} = 142.849$, df = 3, $p = 0.000$).

The age-stage specific survival rate ($S_{xj}$) and age-stage specific fecundity ($m_x$) rates of *A. craccivora* under different temperatures are given graphically in Fig. 1. Here, the first nymph mortality and survival rates up to age x for the adult female and developmental time of *A. craccivora* under different temperatures can be seen approximately. In this graph, the age-stage specific survival rate ($S_{xj}$) curve represents the probability of survival of individuals at a certain age or developmental stage (egg, nymph, pupa, and adult), in other words, ($S_{xj}$) the survival rate of "$j$" individuals at "$x$" age or stage. According to Fig. 1, the temperature at which the pre-adult development curves of *A. craccivora* are higher and more horizontal for a longer time indicates that the survival rate is higher at that temperature. Accordingly, it is seen that the curves decrease more rapidly at 27.5 °C and 30 °C compared to other temperatures, and the mortality rate is higher here. The ($m_x$) curve presented in Fig. 1 expresses the fertility of female individuals at a certain age or period at "x" age or period. The fertility rate of *A. craccivora* changes under different temperature conditions, and the fertility potential is higher at 25 °C, where the ($m_x$) curve is at its highest point, compared with other temperature values, and the fertility rate decreases at 22.5 °C and 30 °C, where the ($m_x$) curve is at its lowest.

In this study, data on some biological properties of *A. craccivora* at different temperatures are given in Table 3. Accordingly, the intrinsic rate of increase ($r$) value shows the growth rate of the population per unit time. The highest value was 0.36 (day$^{-1}$) at 25 °C, while the lowest value was 0.16 day$^{-1}$ at 22.5 °C. The net reproductive rate ($R_0$) value shows the average number of female offspring produced by a female individual throughout her life, and the larger the value, the more the population is transferred to the next generation. In the study, the highest ($R_0$) value was 63.67 (female/female) at 25 °C, whereas the lowest value was 8.18 female/female at 30 °C. The finite rate of increase ($\lambda$) value expresses the growth rate of the population per unit time, and the larger the value,

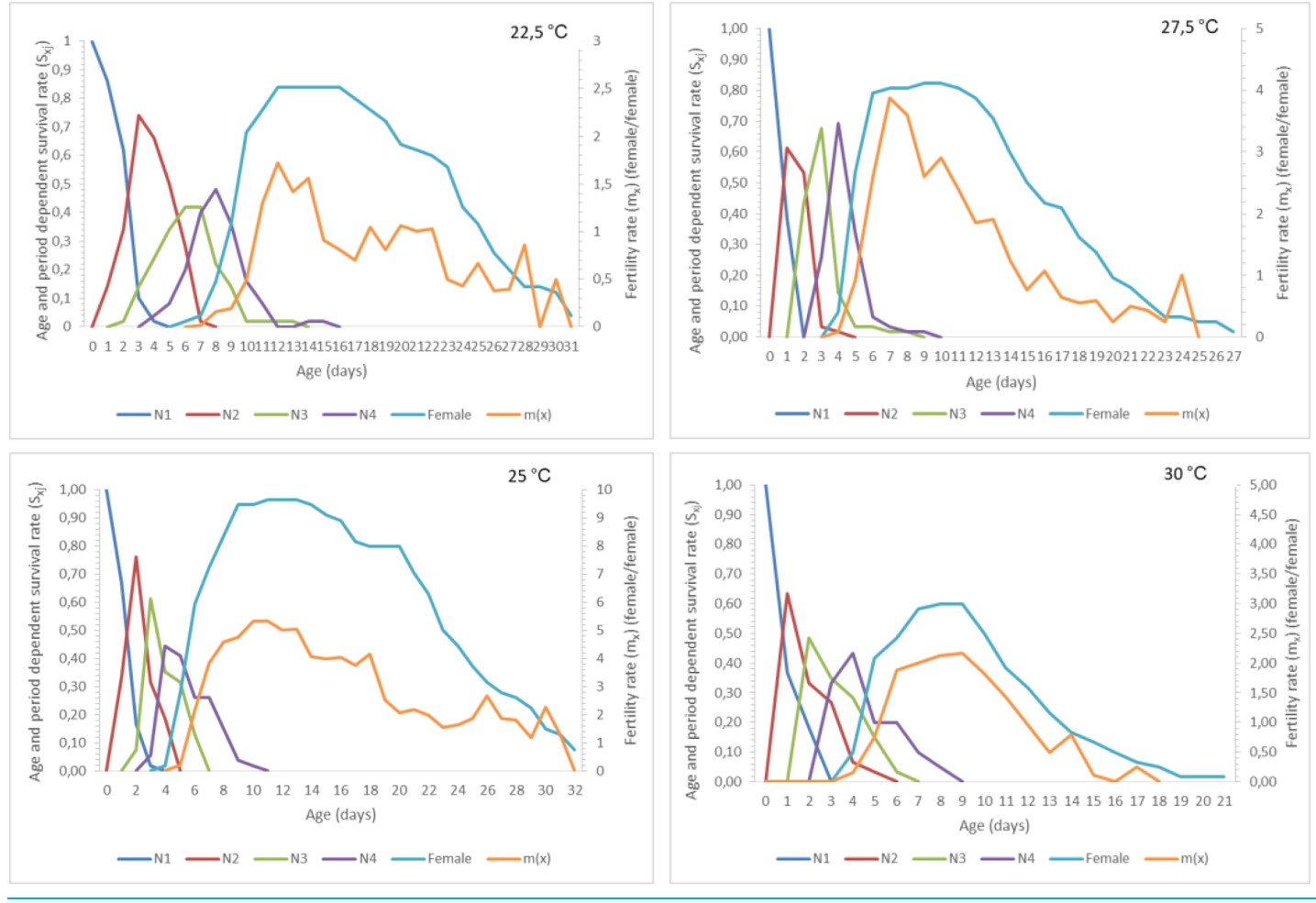

**Figure 1** Age-stage specific survival rate ($S_{xj}$) and age-specific fecundity ($m_x$) of *Aphis craccivora*.

the faster the population grows. In Table 2, the highest ($\lambda$) value was 1.43 day$^{-1}$ at 25 °C, followed by 1.39 day$^{-1}$ at 27.5 °C. The lowest ($\lambda$) value was determined as 1.17 (day$^{-1}$) at 22.5 °C. The gross reproductive rate (GRR) value, which shows the reproductive capacity, indicates the total number of offspring produced by a female individual throughout her life in the study; the highest (GRR) value was 80.88 at 25 °C, whereas the lowest value was 14.97 at 30 °C. The mean generation period (T) value shows the time it takes for one generation to pass to the next, and the shorter it is, the faster the population changes generations. According to Table 3, the lowest (T) value was 8.63 days at 30 °C, while the highest value was 15.56 days at 22.5 °C. The doubling time (DT) value, which indicates how fast the population grows, is again shorter, meaning that the population grows faster. According to Table 3, the lowest (DT) value was 1.93 days at 25 °C, while the highest value was 4.3 days at 22.5 °C.

According to the data obtained here, it is understood that *A. craccivora* has the highest net reproductive power ($R_0$) and finite rate of increase ($\lambda$) values and the lowest doubling

**Table 3 Population parameters of *Aphis craccivora* reared on lentil plant at four constant temperature.**

| Temperature (°C) | 22.5 °C | 25 °C | 27.5 °C | 30 °C |
|---|---|---|---|---|
| | Value ($n = 31$) | Value ($n = 32$) | Value ($n = 27$) | Value ($n = 21$) |
| The intrinsic rate of increase, $r$ (day$^{-1}$) | 0.16 | 0.36 | 0.33 | 0.24 |
| The net reproductive rate, R0 (female/female) | 12.28 | 63.67 | 21.26 | 8.18 |
| The finite rate of increase, $\lambda$ (day$^{-1}$) | 1.17 | 1.43 | 1.39 | 1.28 |
| The gross reproductive rate, (GRR) | 17.97 | 80.88 | 30 | 14.97 |
| Average fertilization period, T (day) | 15.56 | 11.55 | 9.36 | 8.63 |
| Population doubling time, DT (day) | 4.3 | 1.93 | 2.12 | 2.85 |

time (DT) values at 25 °C, and that the population grows and reproduces faster at this temperature. Similarly, at 22.5 °C and 30 °C, the lowest ($R_0$) and ($\lambda$) values, while the highest (T) and (DT) values, show that the population grows and reproduces slowly at these temperatures.

In a similar study conducted on the bean plant, *Dona & Satar (2024)* conducted experiments on *A. craccivora* at different temperatures (16, 20, 24 and 28 °C) and determined the mean generation time (T) as 22.287, 13.294, 13.174 and 10.191 days, respectively, the net reproductive power ($R_0$) as 32.175, 40.850, 46.975 and 26.925 female/female, respectively, and the intrinsic rate of increase ($r$) as 0.17, 0.321, 0.352 and 0.367 (day$^{-1}$), respectively. In this study, the highest (T) value was obtained at 15.56 and 22.5 °C, ($R_0$) value was obtained at 63.67 and 25 °C, and although the internal increase rate ($r$) value in both studies showed similar values, the highest ($r$) value was obtained at 0.36 and again at 25 °C in this study. It is thought that the different results obtained in the findings regarding some life parameters of *A. craccivora* at different temperatures in these studies are due to the different temperature values and nutrients used in the studies.

The rates of age-specific survival ($l_x$), age-specific fecundity ($m_x$), and age-specific fertility of *A. craccivora* under different temperatures are presented in Fig. 2. The age-specific survival rate ($l_x$) gives the probability of survival of individuals at a certain age and is used to determine the age-related mortality rate of the population. According to Fig. 2, the ($l_x$) curve is more horizontal at 25 °C compared to other temperatures, and the survival rate is higher here, whereas the curve becomes steeper at 30 °C, indicating that the mortality rate is higher. The average number of offspring of female individuals at a certain age, and the ($m_x$) curve for fertility, shows the age-related reproductive potential of the population. In Fig. 2, the peak point of the ($m_x$) curve is the highest at 25 °C than at other temperatures, indicating that the fertility potential is higher, while the ($m_x$) curve appears late at 22.5 °C, and because the peak point is lower, reproduction is low. When Fig. 2 is examined for the ($l_xm_x$) curve used to determine the net reproduction rate ($R_0$), the ($l_xm_x$) curve has the largest area at 25 °C, indicating that the population's growth potential is higher.
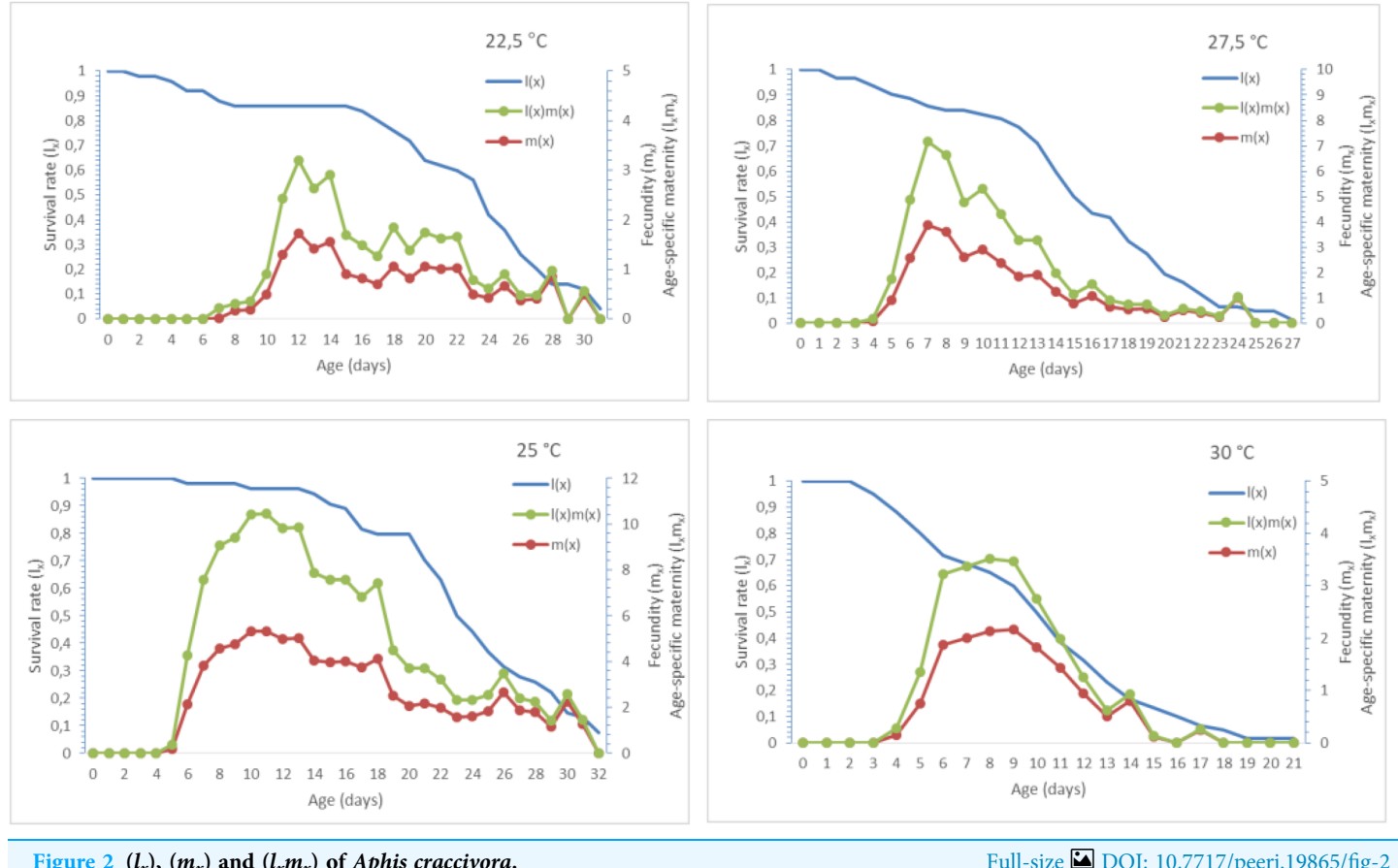

**Figure 2** $(l_x)$, $(m_x)$ and $(l_x m_x)$ of *Aphis craccivora*.

## CONCLUSION

This study examined the life history parameters of *A. craccivora* on red lentil plants under controlled laboratory conditions at various temperatures (22.5, 25, 27.5, and 30 °C). The nymphal development periods were found to be 26.04, 17.24, 23.98, and 26.74 degree-days at these temperatures, respectively. The longest developmental time for *A. craccivora* was observed at 22.5 °C, while the shortest occurred at 27 °C. Regarding adult longevity, the longest average lifespan was 17.24 days at 25 °C, and the shortest was 6.83 days at 30 °C. The differences in adult longevity across the temperature range were statistically significant. The study also showed significant temperature-dependent changes in productivity. The highest productivity (62.74 offspring per female) was observed at 25 °C. This decreased to 23.76 at 27.5 °C, with the lowest productivity values at 30 °C (11.28) and 22.5 °C (13.74), which were statistically grouped. Overall, the shortest developmental time, the longest lifespan, and the highest productivity were all recorded at 25 °C.

Furthermore, the age-stage specific fecundity (mx) rates of *A. craccivora* varied with temperature, peaking at 25 °C and indicating a higher reproductive potential compared to other temperatures. The highest intrinsic rate of increase (r) was 0.36 day$^{-1}$ at 25 °C, while the lowest was 0.16 day$^{-1}$ at 22.5 °C. Similarly, the highest net reproductive rate (R$_0$) was 63.67 females/female at 25 °C, whereas the lowest was 8.18 females/female at 30 °C. The

highest finite rate of increase ($\lambda$), which reflects the population growth rate per unit time, was 1.43 day$^{-1}$ at 25 °C, with the lowest at 1.17 day$^{-1}$ at 22.5 °C. The highest gross reproductive rate (GRR) was 80.88 at 25 °C, and the lowest was 14.97 at 30 °C. Conversely, the shortest mean generation time (T) was 8.63 days at 30 °C, and the longest was 15.56 days at 22.5 °C. The shortest doubling time (DT) was 1.93 days at 25 °C, while the longest was 4.3 days at 22.5 °C. In summary, this study highlights that 25 °C provides the most favorable conditions for *A. craccivora*'s population growth and development, characterized by a shorter preimaginal phase, longer adult longevity, and optimal reproductive and population increase parameters. These findings are crucial for understanding the population dynamics of *A. craccivora* and for developing effective pest management strategies.

## ACKNOWLEDGEMENTS

The diagnosis of Aphis craccivora Koch (Hemiptera: Aphididae) was made by Prof. Dr. Selime ÖLMEZ BAYHAN (Dicle University, Faculty of Agriculture, Department of Plant Protection, Diyarbakır). This study was conducted in the laboratory and climate chambers using the facilities of the Diyarbakır Plant Protection Research Institute Directorate, affiliated with the General Directorate of Agricultural Research and Policies of the Ministry of Agriculture and Forestry of the Republic of Turkey. We would like to thank Prof. Dr. Hsin Chi for providing program use, information, and documentation support in life cycle studies.

### Funding

The authors received no funding for this work.

### Competing Interests

The authors declare that they have no competing interests.

### Author Contributions

- Muhlis Sezgin conceived and designed the experiments, performed the experiments, analyzed the data, prepared figures and/or tables, authored or reviewed drafts of the article, and approved the final draft.
- Merve Akyıldız conceived and designed the experiments, performed the experiments, authored or reviewed drafts of the article, and approved the final draft.
- Selime Olmez Bayhan analyzed the data, authored or reviewed drafts of the article, and approved the final draft.
- Erol Bayhan analyzed the data, prepared figures and/or tables, and approved the final draft.

### Data Availability

Raw data is available in the Supplemental Files.

## Supplemental Information

Supplemental information for this article can be found online at http://dx.doi.org/10.7717/peerj.19865#supplemental-information.

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
