# Peer review of "Effect of temperatures on some biological parameters of Aphis craccivora Koch (Hemiptera: Aphididae) on lentil"

_PeerJ, doi:10.7717/peerj.19865_

## Round 0.1 · original submission · Major Revisions

· Academic Editor

Major Revisions

Dear Dr. Sezgin, I ask you to take into account all the comments of the reviewers. The manuscript needs to be adapted to international standards of scientific publications (statistical processing, tables, diagrams). I hope that the new version of this article will allow the reviewers to approve it for publication.

**Language Note:** The review process has identified that the English language must be improved. PeerJ can provide language editing services - please contact us at [email protected] for pricing (be sure to provide your manuscript number and title). Alternatively, you should make your own arrangements to improve the language quality and provide details in your response letter. – PeerJ Staff

Reviewer 1 ·

Basic reporting

The topic of the study is relevant and has considerable practical and theoretical importance. The authors have conducted an extensive literature review, which allowed them to substantiate the research topic. However, the way the topic is currently framed gives it a local character. The authors should revise the formulation to better highlight issues and methods that would be of interest to a broader international research audience beyond Turkiye.

Line 60: “Lentils (Lens culinaris) are an important source of plant protein worldwide especially in Turkiye with their high protein content” – this sentence is somewhat tautological and stylistically leans more toward popular science than formal scientific writing.

Line 61: The introductory clause “At the same time, …” is not very appropriate, as the preceding sentence does not refer to any specific temporal context.

Lines 72–80: References to relevant literature are needed.

Line 87: The sentence “It is well known that there are many studies on the biology of A. craccivora” is essentially uninformative.

Line 87: The statement “However, there are no studies on the biological parameters of this aphid on the lentil plant” is questionable—an express literature search contradicts this claim.

Experimental design

The methodology is well documented and clearly presented. The experimental protocols and the amount of collected data are sufficient to support the conclusions. The methods are reproducible, and the statistical procedures are appropriate for the stated objectives and align with modern standards in research on the biological traits of pests.

Validity of the findings

The results are presented clearly and comprehensively. However, the article lacks several crucial components. It would be highly valuable if the authors included a critical assessment of the limitations of their methodology and proposed possible improvements. Additionally, the paper would benefit from a discussion of the prospects for future research and the practical implementation of the findings. Importantly, the article should conclude with a summary of the main conclusions.

Table 1: The table should include definitions of the abbreviations (X, Y), the sample size (N), and—most importantly—the statistical significance of the regression coefficients. A high explanatory power of the model may result from the constant term, whereas the primary objective of the model is to assess the slope of the relationship. Furthermore, it is essential to specify the temperature range for which the reported relationships are valid.

Line 194: “In particular, the r² value is quite high (between 0.73–0.9784 and 0.9084 in total), which confirms the relationship between temperature and development rate, and the high r² values, especially in the third and fourth periods, show that the development rate in these periods is very dependent on temperature” – formally, r² measures the quality of the model fit, not the strength of the relationship between predictor and response per se. As the sample size increases, r² may decrease while the statistical significance increases, although the actual relationship between predictor and response remains unchanged.

Table 3: Population parameters of Aphis craccivora reared on lentil plant at four constant temperatures – it would be better to report these parameters along with measures of their variability.

Reviewer 2 ·

Basic reporting

1.The use of the "F1rat87" lentil variety is justified, but no details are provided on its secondary metabolite content (e.g., tannins, saponins), which may confound aphid responses.
Revision Suggestions: Include phytochemical analyses or reference prior studies on lentil-aphid interactions.
2.In-text citations must align with the reference list. The manuscript repeatedly cites non-public data labeled as "Anonymous" (e.g., "Anonymous 2023a," "Anonymous 2023b") without specifying the source institution, report title, or accessibility details (e.g., internal database, unpublished agency document).
Revision Suggestions: Source Identification: Provide full citations for all "Anonymous" references, including: The name of the institution/organization responsible for the data. Report/document titles (if applicable). Accession numbers or persistent identifiers (e.g., DOI, URL) for traceability.
Reference List Update: Remove any citations to unverifiable "Anonymous" sources from the reference list unless they are formally published gray literature (e.g., technical reports with institutional attribution).

Experimental design

1.In the part of Plant Source: [Rationale Missing] On what basis was the F1rat87 red lentil cultivation environment selected? No criteria (e.g., temperature/humidity thresholds, soil properties) are stated.
[Equipment Details Incomplete]: Photoperiod equipment: Model/brand of greenhouse humidity control devices (e.g., humidifier accuracy) is unspecified. Light sources: Type (e.g., LED vs. fluorescent) and spectral specifications are not provided.
In the part of Insect Source: [Identification Records Absent]: Wild-collected aphids lack morphological/molecular identification records (e.g., no voucher specimen repository or GenBank accession numbers).
Revision Suggestions: Clarify the environmental selection criteria for F1rat87 cultivation. Specify full technical details of photoperiod/humidity control equipment (brand, model, accuracy). Include aphid identification protocols and repository information for traceability.

Validity of the findings

1.The temperature descriptions must be consistent throughout the manuscript. The abstract mentions 27°C, but this temperature is absent in the Methods section.
Revision Suggestions:
Standardization of Temperature Parameters: Unify all temperature values to reflect the actual experimental temperatures tested (e.g., replace "27°C" in the abstract with the exact temperature gradient used in the Methods, such as 22.5°C, 25°C, 27.5°C, or 30°C).
Data Re-analysis Request: Re-analyze the developmental stage data (e.g., nymphal duration, adult longevity) according to the real temperature gradient implemented in the experiments (i.e., 22.5°C, 25°C, 27.5°C, 30°C), ensuring all conclusions align with the tested conditions.

·

Basic reporting

Studying the temperature regimes of aphids is an important area of modern science. These data allow us to identify the most vulnerable periods in the life cycle of aphids and develop control methods that are most effective in specific conditions. Therefore, the topic of the article under review is very relevant. However, there are a number of serious comments on the design of the manuscript.
When mentioning an animal organism for the first time in the abstract and text of the manuscript, it is necessary to indicate the full generic and species name, the author's surname and the year of description of the species in accordance with the International Code of Zoological Nomenclature.
Rephrase the sentence (lines 60–61): "Lentils (Lens culinaris) are an important source of plant protein worldwide especially in Türkiye with their high protein content".
I recommend shortening part of the text in the "Introduction" section (phrases that are not directly related to the topic of the manuscript, lines 61–70) to one or two sentences.

Experimental design

The Introduction section contains very few references to literature and is insufficient in volume. I recommend that the authors familiarize themselves with the literature on this topic in more detail. The authors claim that there are no studies on the topic of the article (lines 87–89). This is an incorrect statement. There are publications, for example: “Ahlawat, D. S., Verma, T., & Yadav, R. (2022). Population dynamics of major insect-pests of lentil and correlation with abiotic factors. Journal of Food Legumes, 35(1), 47–50. https://doi.org/10.59797/journaloffoodlegumes.v35i1.361” and others. Perhaps the authors meant that similar studies have not been conducted in Turkey?
What is the novelty of your research? Add a few sentences to the text.
Lines 121–128. The authors describe the laboratory studies and indicate the repetition of the experiments. It is incorrect to cite literary sources in this case. Are these your studies or those of other authors?
The Materials and Methods section is disproportionately large in volume. I recommend shortening it.

Validity of the findings

Rephrase the sentences or combine the information (lines 187–191). It is incorrect to refer to Table 1 in every sentence.
It is better to replace the phrase “biological parameters” of Aphis craccivora with “biological characteristics”.
I recommend that the authors separate the results and discussions into separate sections. In the Results section, citing literary sources is not allowed.
Line 187 “pre-adult stages” should be replaced with “preimaginal phases of development”.
The generic Latin name of the studied aphid species should be abbreviated in the abstract and the text of the manuscript after the first mention (A. craccivora).
Describe the limitations of the study and add to the Discussion section.
In the Results and Discussion section, lines 297–300, the authors cite literary data on the effect of different temperature regimes on the development of preimaginal phases of Aphis craccivora. However, they do not indicate which plants the studies were conducted on. It is necessary to add.
The Conclusions section is missing. Add.
The bibliography is missing some sources cited in the text (Lu and Kuo, 2008) (line 127).

Additional comments

No comments.

---

## Round 0.2 · accepted · Accept

· Academic Editor

Accept

Dear Dr. Sezgin, I am pleased to inform you that your article has been accepted for publication. I hope that you will continue to study plant pests and send articles of the same high quality to our journal in the future.

Reviewer 1 ·

Basic reporting

The authors have implemented all the recommendations of the reviewer. The quality of the manuscript has been significantly improved. I recommend the article for publication.

Experimental design

The authors have implemented all the recommendations of the reviewer. The quality of the manuscript has been significantly improved. I recommend the article for publication.

Validity of the findings

The authors have implemented all the recommendations of the reviewer. The quality of the manuscript has been significantly improved. I recommend the article for publication.

·

Basic reporting

The authors of the manuscript took into account all my previous comments. They made changes to the text of the article. I believe that the manuscript can be recommended for publication.

Experimental design

The authors of the manuscript took into account all my previous comments. They made changes to the text of the article. I believe that the manuscript can be recommended for publication.

Validity of the findings

The authors of the manuscript took into account all my previous comments. They made changes to the text of the article. I believe that the manuscript can be recommended for publication.

Additional comments

No comments.